# Association of Plasma Lipopolysaccharide-Binding Protein Concentration with Dietary Factors, Gut Microbiota, and Health Status in the Japanese General Adult Population: A Cross-Sectional Study

**DOI:** 10.3390/metabo13020250

**Published:** 2023-02-09

**Authors:** Nobuo Fuke, Takahiro Yamashita, Sunao Shimizu, Mai Matsumoto, Kaori Sawada, Songee Jung, Itoyo Tokuda, Mina Misawa, Shigenori Suzuki, Yusuke Ushida, Tatsuya Mikami, Ken Itoh, Hiroyuki Suganuma

**Affiliations:** 1Innovation Division, KAGOME Co., Ltd., 17 Nishitomiyama, Nasushiobara 329-2762, Tochigi, Japan; 2Department of Vegetable Life Science, Hirosaki University Graduate School of Medicine, 5 Zaifu-cho, Hirosaki 036-8562, Aomori, Japan; 3Innovation Center for Health Promotion, Hirosaki University Graduate School of Medicine, 5 Zaifu-cho, Hirosaki 036-8562, Aomori, Japan; 4Department of Digital Nutrition and Health Sciences, Hirosaki University Graduate School of Medicine, 5 Zaifu-cho, Hirosaki 036-8562, Aomori, Japan; 5Center of Innovation Research Initiatives Organization, Hirosaki University, 5 Zaifu-cho, Hirosaki 036-8562, Aomori, Japan; 6Department of Stress Response Science, Center for Advanced Medical Research, Hirosaki University Graduate School of Medicine, 5 Zaifu-cho, Hirosaki 036-8562, Aomori, Japan

**Keywords:** lipopolysaccharide-binding protein, metabolic endotoxemia, diet, health status, gut microbiota, cross-sectional study

## Abstract

The influx of intestinal bacteria-derived lipopolysaccharide (LPS) into the blood has attracted attention as a cause of diseases. The aim of this study is investigating the associations between the influx of LPS, dietary factors, gut microbiota, and health status in the general adult population. Food/nutrient intake, gut microbiota, health status and plasma LPS-binding protein (LBP; LPS exposure indicator) were measured in 896 residents (58.1% female, mean age 54.7 years) of the rural Iwaki district of Japan, and each correlation was analyzed. As the results, plasma LBP concentration correlated with physical (right/left arms’ muscle mass [*β* = −0.02, −0.03]), renal (plasma renin activity [*β* = 0.27], urine albumin creatinine ratio [*β* = 0.50]), adrenal cortical (cortisol [*β* = 0.14]), and thyroid function (free thyroxine [*β* = 0.05]), iron metabolism (serum iron [*β* = −0.14]), and markers of lifestyle-related diseases (all *Q*s < 0.20). Plasma LBP concentration were mainly negatively correlated with vegetables/their nutrients intake (all *βs* ≤ −0.004, *Q*s < 0.20). Plasma LBP concentration was positively correlated with the proportion of *Prevotella* (*β* = 0.32), *Megamonas* (*β* = 0.56), and *Streptococcus* (*β* = 0.65); and negatively correlated with *Roseburia* (*β* = −0.57) (all *Q*s < 0.20). Dietary factors correlated with plasma LBP concentration correlated with positively (all *β*s ≥ 0.07) or negatively (all *β*s ≤ −0.07) the proportion of these bacteria (all *Q*s < 0.20). Our results suggested that plasma LBP concentration in the Japanese general adult population was associated with various health issues, and that dietary habit was associated with plasma LBP concentration in relation to the intestinal bacteria.

## 1. Introduction

Lipopolysaccharides (LPS) is a molecule composed of lipids and polysaccharides that is part of the outer membrane of Gram-negative bacteria. LPS derived from intestinal bacteria flow into the blood in the association with dietary habits and gut microbiota, and induces inflammatory reactions through Toll-like receptor 4 [1]. Chronic inflammation is thought to be associated with the development of various age-related diseases [2]. Based on these findings, the prevention and improvement of health issues by reducing blood LPS concentrations, through dietary intervention, has been attempted [1].

There are three issues in this research area. Firstly, the correlation between the influx of LPS and deterioration of health status has only been elucidated for some limited diseases. To date, correlations between blood LPS concentration and obesity [3], insulin resistance [4], diabetes [5], non-alcoholic fatty liver disease [6], pancreatitis [7], amyotrophic lateral sclerosis [8], and Alzheimer’s disease [8] have been reported in case-control studies. However, as noted above, LPS may be a factor in a wide range of health issues by inducing chronic inflammation [2]. Therefore, the correlation between the various health issues in the general adult population and blood LPS should be comprehensively elucidated.

Second, the dietary factors related to influx of LPS have not been fully clarified. It has been reported that lipid intake induces the influx of LPS into the blood, and probiotics, prebiotics, and polyphenols reduce blood LPS concentration [1]. These previous studies were mainly intervention studies focusing on the effectiveness of specific food materials and ingredients. However, we consume a wide variety of foods and their constituents in our daily diet, which may also influence the influx of LPS. Regarding the relationship between daily dietary intake and LPS influx, Röytiö et al. evaluated the relationship between blood LPS concentrations and intakes of energy, three dietary fibers, protein, three fatty acids, five vitamins and nine minerals in 88 overweight pregnant women, but none of the nutrients correlated with blood LPS concentrations [9]. Ahola et al. also found that three dietary patterns (‘fish’, ‘healthy snack’ and ‘modern’) were negatively associated with blood LPS levels in 668 patients with type 1 diabetes [10]. However, the relationship between individual foods and nutrients and LPS influx is not fully understood. The two epidemiological studies described above were conducted in overweight pregnant women and patients with type 1 diabetes, and there is concern that the dietary habits of the study population may be biased. Therefore, the association between individual foods and nutrients we consume in our daily lives and the influx of LPS are needed to fully investigate in general adult population.

Third, the comprehensive relationships between food and nutrient intake, gut microbiota, LPS influx and health status are not well understood. Previous intervention studies with specific dietary factors have evaluated the changes in gut microbiota, blood LPS concentrations, and specific health status, associated with the intervention [1]. However, few epidemiological studies have comprehensively evaluated all relationship between dietary factors, gut microbiota, blood LPS influx, and health status, in daily life within the same study. Luthold et al. reported that in 150 healthy subjects, blood 25-hydroxyvitamin D levels were negatively associated with blood LPS levels, and blood 25-hydroxyvitamin D levels were negatively associated with the composition of *Coprococcus* and *Bifidobacterium* in gut microbiota [11]. However, no association was found between these two gut bacteria and blood LPS levels, and the involvement of gut bacteria in the association between vitamin D intake and lower blood LPS levels has not been revealed. In the above-mentioned study by Röytiö et al., a positive association between dietary fiber intake and the composition of Firmicutes and Barnciellaceae, a positive association between vitamin A and *β*-carotene intake and the composition of Firmicutes, and a negative association between fat intake and the composition of Barnciellaceae were found. However, no association was found between these intestinal bacteria and blood LPS concentrations. 

To address these three issues, we conducted a cross-sectional study in the general adult population in which the participants’ dietary intake, LPS influx, gut microbiota, and health status were examined, and each correlation were analyzed. 

## 2. Materials and Methods

### 2.1. Ethics Approval and Consent to Participate

This study was conducted according to the guidelines of the Declaration of Helsinki 2013, and all procedures involving human subjects were approved by the ethics boards of Hirosaki University Graduate School of Medicine (2017-026) and the ethics committees of KAGOME Co., Ltd. (2017-R06). Written informed consent was obtained from all subjects.

### 2.2. Study Design and Population

The subjects of this cross-sectional study were participants of the Iwaki Health Promotion Project conducted at the Iwaki district of Hirosaki city, Aomori prefecture between May 27th and June 5th, 2017. The Iwaki Health Promotion Project (UMIN000040459) is an annual health survey conducted by Hirosaki University with the aim of preventing lifestyle-related diseases and extending healthy life expectancy for residents of the Iwaki district [12,13,14]. Approximately 1000 out of 12,000 residents participate in the project every year. A feature of this health survey is that it simultaneously surveys dietary habits, gut microbiota, and a large amount of health status. Therefore, it is useful for evaluating the relationship between dietary factors, gut microbiota, blood LPS influx, and health status in the general adult population. The subjects were recruited through flyers posted and distributed in the Iwaki district and through the website. The inclusion criteria for the study subjects were male and female aged 20 years or older living in the Iwaki district. A total of 1071 people participated in this survey. Following the exclusion of subjects with one or more missing values in the items used for the analysis, the data of 896 subjects were analyzed.

### 2.3. Collection of Attribute Information

Sex, age, smoking habit (current habitual smoker or not) and drinking habit (current habitual drinker or not) were self-reported by the participants on the questionnaires. Subjects were asked to report their habitual drinking throughout the year by choosing one of three options: never drink (including drinking several times a month at parties), used to drink, or still drink. Subjects who answered, ‘never drink’ and ‘used to drink’ were classified as non-drinkers, while those who answered ‘still drink’ were classified as current habitual drinkers. Subjects were also asked to report their current smoking status by selecting one of three options: never smoked, ever smoked or currently smoking. Of these, those who answered, ‘never smoked’ and ‘ever smoked’ were considered to be a non-smoker, while those who answered ‘currently smoking’ were considered to be a current habitual smoker.

### 2.4. Blood Sampling and Measurement of Plasma LBP Concentrations

LPS is inactivated in a temperature-dependent manner in plasma, and LPS concentration decreases even when plasma is cryopreserved [12]. Therefore, it is difficult to accurately measure LPS concentrations in large epidemiological studies. The LPS-binding protein (LBP) is a protein produced mainly in the liver in response to LPS [13]. A significant positive correlation between serum LBP concentrations and serum LPS concentrations has been reported in humans [14]. Furthermore, there are no reports on decreased LBP concentration due to cryopreservation of plasma or serum. Therefore, plasma or serum LBP concentration is utilized as a surrogate marker for blood LPS concentration in human cohort [15,16,17] and cross-sectional studies [18]. Therefore, in this study, plasma LBP concentrations were measured as an indicator of LPS influx.

Blood samples were obtained from the median cubital vein after an overnight fast. Heparinized plasma was prepared from the blood samples, and LBP concentration was determined for each study subject. A commercially available enzyme-linked immunosorbent assay (ELISA) kit (ALX-850-305-KI0, Enzo Life sciences, Farmingdale, NY, USA) was used to measure plasma LBP concentrations in accordance with the manufacturer’s instructions.

### 2.5. Clinical Markers and Clinical Scores

A total of 100 items were used for the analysis, including health survey-obtained clinical markers relating to obesity, blood pressure, arteriosclerosis, glucose metabolism, lipid metabolism, physical function, eyesight, liver function, renal function, adrenal cortex function, cardiac function, thyroid function, sexual function, inflammation, oxidative stress, hematological test, iron metabolism, electrolyte, allergy, rheumatoid arthritis, pituitary gland function, and asthma/airway inflammation; clinical scores relating to mental health, depression, and cognitive function. Specific measurement items and measurement methods are shown in Appendix A.

In particular, visceral fat levels were determined by firmware embedded in the body composition analyser (Tanita MC190, TANITA Corp., Tokyo, Japan) using a proprietary equation developed by the manufacturer. Levels 9 and below are standard, levels 10–14 are slightly excessive and levels 15 and above indicate excessive visceral fat accumulation. Level 10 corresponds to 100 cm^2^ of visceral fat measured on a computerized tomography scan. Basal metabolic rate levels were also determined by the firmware built into the body composition analyser using a unique formula developed by the manufacturer. Specifically, basal metabolic rate levels were calculated on a scale of 1–16 based on the average and statistical distribution of basal metabolic reference values by age group. Levels 6 and below indicated low fat burning, levels 7–10 were standard and levels 11 and above indicated easy burning. 

Following previous reports, units were not described for homeostasis model assessment-insulin resistance (HOMA-IR) [19], far- and near-sightedness [20].

### 2.6. Dietary Factors

Food and nutrition intake were evaluated with a brief-type self-administered diet history questionnaire (BDHQ) [21,22,23]. The BDHQ is a four-page self-administered questionnaire [24]. The questionnaire assesses dietary habits over the previous month and consists of five sections: (i) frequency of consumption of 46 food and non-alcoholic beverage items, (ii) daily intake of rice (including types of rice) and miso soup, (iii) frequency and amount of alcoholic beverages consumed, (iv) usual cooking methods, and (v) general dietary behavior. Many of the foods and beverages were selected from items commonly consumed in Japan, and some were added using a food list from the National Health and Nutrition Survey of Japan. Dietary intake estimates for seventy foods and beverages and ninety-nine nutrients were calculated by the developer (EBNJAPAN, Tokyo, Japan) using an ad hoc computer algorithm. The validity of BDHQ estimates of food and nutrient intakes has been validated in previous reports [24,25]. Of the ninety-nine nutrients, energy, protein, fat, carbohydrate, three fatty acids, cholesterol, three dietary fibers, nine minerals, eleven vitamins and alcohol were used as representative nutrients for analysis in this study. All seventy foods were included in the statistical analysis of this study. The concentrations of serum carotenoid (lutein, zeaxanthin, *β*-cryptoxanthin, *α*-carotene, *β*-carotene, lycopene), serum retinol, and serum *α*-tocopherol were measured using high-performance liquid chromatography (HPLC) according to a previous report [26]. We used a C30 carotenoid column and photodiode array detector (Prominence LC-30AD/Nexera X2 SPPD-M30A, SHIMADZU CORPORATION, Kyoto, Japan) for HPLC analysis. Plasma vitamin C concentration was measured using a commercially available measurement kit (ROIK02, SHIMA Laboratories Co, Ltd., Tokyo, Japan) in accordance with the manufacturer’s instructions.

### 2.7. Analysis of Intestinal Bacteria by Next-Generation Sequencing

Analysis of gut microbiota was conducted according to a previous report [27]. Specifically, fecal deoxyribonucleic acid (DNA) was extracted from the feces of the study subjects and the V3-V4 region of the 16S ribosomal ribonucleic acid (rRNA) sequence was amplified with a polymerase chain reaction using a universal primer. An Illumina MiSeq sequencing system (Illumina, San Diego, CA, USA) and MiSeq Reagent Kit 3 (Illumina) was used to analyze this sequence. Part of the base sequence (approximately 380–430 bp) of the analyzed 16S rRNA sequence was clustered using VSEARCH (v. 2.4.3), with homology of at least 97% set as a criterion. Clusters with a confidence level of less than 0.8 were grouped as an unclassified taxon. We used an RDP classifier (commit hash: 701e229dde7cbe53d4261301e23459d91615999d) for cluster classification. The relative abundance of gut bacteria was calculated by dividing the number of reads for each bacterium by the total number of reads. Bacteria with an average relative occupancy rate of less than 1% were excluded from the analysis because we could not reject the possibility of noise.

### 2.8. Statistics

Background data of subjects were presented as frequency (percentage) for categorical variables; and mean ± standard deviation (SD), median, and interquartile range (IQR) for continuous variables. The normality of data was assessed by the Shapiro-Wilk test. As described in ‘Results’ section, all continuous value data, except the hematocrit were not normally distributed. Thus, logarithmic transformed values of plasma LBP concentrations, clinical marker values (excluding hematocrit), and blood antioxidant concentrations were used for statistical analysis. Eosinophils, basophils, clinical scores, and food and nutrient intakes calculated from the BDHQ were used as raw values for statistical analysis because the data contain zeros and cannot be converted to logarithms. Additionally, although the data of the intestinal bacterial also included 0, a pseudo-count of 1 was added to all bacteria as previously described [28], after which a logarithmic transformation was performed. 

The Mann-Whitney *U* test was used to compare age and health status in male and female. Fisher’s exact test was used to compare the proportion of current smokers and drinkers in male and female.

In the analysis of correlation between age and plasma LBP concentration, the study subjects were stratified into the following six categories according to age: <30 years, 30–39 years, 40–49 years, 50–59 years, 60–69 years, 70–79 years, and ≥80 years. The Jonckheere-Terpstra test was used to evaluate the correlation. In the analysis of correlation between body mass index (BMI) and plasma LBP concentration, the study subjects were stratified into the three BMI categories in accordance with the criteria developed by world health organization: <18.5 kg/m^2^, 18.5–24.9 kg/m^2^, and ≥25.0 kg/m^2^, and the difference in plasma LBP concentration between each group was evaluated using the Steel-Dwass test. Differences in plasma LBP concentration according to sex, smoking habit (current habitual smoker or not), and drinking habit (current habitual drinker or not) were evaluated using the Mann-Whitney *U* test. 

Multiple regression analysis adjusted for confounding variables was used to evaluate the correlations between plasma LBP concentration and clinical markers and clinical scores, dietary factors and plasma LBP concentration, gut microbiota composition ratio and plasma LBP concentration, and dietary factors and gut microbiota composition ratio. Age, sex, BMI, smoking habit (current habitual smoker or not), and drinking habit (current habitual drinker or not) were used as confounding variables. Energy intake was also used as a confounding variable in the multiple regression analysis with dietary factors as the explanatory variable. Additionally, the intake of each dietary factor was adjusted by the residual method [29] for analysis. For the results of multiple regression analysis, the *Q* value was calculated by the Storey method in order to control false discovery rate. In a multiple regression analysis assessing the association between plasma LBP concentrations and food or nutrient intakes, *β* values were first calculated as indicators of the relationship between intakes of 1 g, 1 mg or 1 μg of a food or nutrient and plasma LBP concentrations. However, in general, foods and nutrients are rarely consumed in amounts as small as 1 g, 1 mg or even 1 μg. Therefore, to facilitate understanding of the relationship between food and nutrient intakes and plasma LBP concentrations, *β* values were recalculated in terms of intakes of 100 g, 100 mg or even 100 μg of food or nutrient (as shown in Table 4). The *β* conversion was performed by multiplying the *β* values obtained from the multiple regression analysis by 100. R ver. 4.0.1 was used for statistical analysis, and *p* < 0.05 and *Q* < 0.20 were considered statistically significant.

## 3. Results

### 3.1. Characteristics of Participants

The sex, age, smoking habit, drinking habit, and health status of the study subjects are shown in Table 1 and Appendix A; the blood antioxidant concentration, food intake, and nutrition intake are shown in Appendix A; and the composition ratio of each gut bacterium is shown in Appendix A. The results of the Shapiro-Wilk test showed that all continuous value data, except the hematocrit were, not normally distributed (*p* < 0.05). Participants were slightly more female (58.1%), with an average age of 54.7 years and 14.3% were current smokers, and 48.5% were current drinkers. In Japan, a BMI of 25 kg/m^2^ or higher is considered obese, and the study subjects were distributed across a wide BMI range, from normal weight to obese. When comparing the characteristics and health status of male and female, there were no significant differences in age between male and female (*p* = 0.5). The proportion of current smoker (*p* = 8 × 10^−13^) and drinker (*p* = 5 × 10^−27^) was significantly higher among male than among female. Male had significantly higher levels of BMI (*p* = 2 × 10^−11^), abdominal circumference (*p* = 2 × 10^−25^), systolic (*p* = 6 × 10^−6^) and diastolic (*p* = 2 × 10^−6^) blood pressure, brachial-ankle pulse wave velocity (baPWV) (*p* = 7 × 10^−6^), blood glucose (*p* = 5 × 10^−9^) and triglyceride (*p* = 2 × 10^−22^) than female. Male had significantly lower levels of total cholesterol (*p* = 0.008) and high-density lipoprotein-cholesterol (HDL-c) (*p* = 4 × 10^−25^) than female. They also tended to have lower insulin levels (*p* = 0.07). There were no significant gender differences in HOMA-IR (*p* = 0.9), hemoglobin A1c (HbA1c) (*p* = 0.8) and high-density lipoprotein-cholesterol (LDL-c) values (*p* = 0.2).

### 3.2. Distribution of Plasma LBP Concentration According to Participant Characteristics

The median plasma LBP concentration was 5.66 µg/mL (IQR 4.63–6.70 μg/mL) (Table 2). Plasma LBP concentration increased significantly with increased age and BMI. Additionally, male subjects had significantly higher plasma LBP concentrations than female subjects. Current smokers had significantly higher plasma LBP concentrations than current non-smokers. There were no significant differences in plasma LBP concentrations between current drinkers and current non-drinkers.

### 3.3. Association of Plasma LBP Concentration and Clinical Markers or Clinical Scores

We conducted multiple regression analysis using values of clinical marker or clinical score as the objective variables; plasma LBP concentrations as the explanatory variables; and age, sex, BMI, smoking habit (current habitual smokers or not), and drinking habit (current habitual drinkers or not) as the adjusting factors (Appendix A). Only categories that were significantly associated with plasma LBP concentrations are selected in Table 3. Results showed that plasma LBP concentration was significantly correlated with 38 clinical markers. Specifically, plasma LBP concentration was significantly correlated with factors related to the following: obesity (abdominal circumference, visceral fat level, basal metabolic rate level), blood pressure (systolic blood pressure, diastolic blood pressure), arteriosclerosis (baPWV), glucose metabolism (HOMA-IR, HbA1c, insulin, C-peptide), lipid metabolism (triglyceride, HDL-c), physical function (both arms’ muscle mass), liver function (aspartate aminotransferase [AST], alanine aminotransferase [ALT], γ-glutamyltransferase [γ-GTP], albumin, total bilirubin, blood total protein concentration), renal function (plasma renin activity, urine albumin creatinine ratio), adrenal cortex function (cortisol), thyroid function (free thyroxine [T4]), inflammation (neutrophil, stab neutrophil, segmented neutrophil, lymphocyte, eosinophil, basophil, immunoglobulin [Ig] G, IgA, complement C3 and C4, high-sensitivity C-reactive protein [hs-CRP], interleukin [IL] -6), and iron metabolism (serum iron). Plasma LBP concentration was also significantly correlated with white blood cell.

### 3.4. Association of Plasma LBP Concentration and Dietary Factors

Multiple regression analysis was conducted using plasma LBP concentration as the objective variable; dietary factors as the explanatory variable; and age, sex, BMI, smoking habit (current habitual smoker or not), drinking habit (current habitual drinker or not), and energy intake as adjustment factors (Appendix A). Only categories that were significantly associated with plasma LBP concentrations are selected in Table 4. Results showed that plasma LBP concentration was positively correlated with six dietary factors and negatively correlated with 16 dietary factors. Specifically, plasma LBP concentration was positively correlated with serum retinol concentration, plasma vitamin C concentration, alcohol intake, shochu intake, raw fish intake, and green tea intake. Additionally, plasma LBP concentration was negatively correlated with serum total carotenoid concentration, serum *β*-cryptoxanthin concentration, serum *β*-carotene concentration, fat intake, monounsaturated fatty acid (MUFA) intake, polyunsaturated fatty acid (PUFA) intake, dietary fiber intake (total dietary fiber, soluble dietary fiber, insoluble dietary fiber), potassium intake, vitamin K intake, vitamin B1 intake, pantothenic acid intake, cabbage/Chinese cabbage intake, Japanese radish/turnip intake, and tomato intake.

**Table 4 metabolites-13-00250-t004:** Association between plasma LBP concentration and dietary factors ^1,2^ (Only categories that were significantly associated with plasma LBP concentrations are selected).

Category	Variable	*β*	*Q* Value
Blood antioxidant	Total carotenoids (μg/mL) ^3^	−0.05	0.19 *
	Lutein (μg/mL)	−0.04	0.33
	Zeaxanthin (μg/mL)	−0.02	0.55
	*β*-Cryptoxanthin (μg/mL)	−0.05	0.12 *
	*α*-Carotene (μg/mL)	−0.01	0.71
	*β*-Carotene (μg/mL)	−0.04	0.08 *
	Lycopene (μg/mL)	−0.02	0.40
	Retinol (μg/mL)	0.09	0.08 *
	Vitamin C (μg/mL)	0.06	0.16 *
	*α*-Tocopherol (μg/mL)	0.06	0.36
Nutrition intake	Protein (100 g)	−0.11	0.36
	Fat (100 g)	−0.19	0.15 *
	SFA (100 g)	−0.48	0.26
	MUFA (100 g)	−0.47	0.17 *
	PUFA (100 g)	−0.67	0.19 *
	Cholesterol (100 mg)	−0.01	0.43
	Carbohydrate (100 g)	0.00	0.80
	Total dietary fiber (100 g)	−0.75	0.11 *
	Soluble dietary fiber (100 g)	−2.62	0.10 *
	Insoluble dietary fiber (100 g)	−1.09	0.11 *
	Sodium (100 mg)	0.00	0.69
	Potassium (100 mg)	−0.004	0.17 *
	Calcium (100 mg)	−0.01	0.42
	Magnesium (100 mg)	−0.03	0.31
	Phosphorus (100 mg)	−0.01	0.36
	Iron (100 mg)	−0.86	0.32
	Zinc (100 mg)	−1.63	0.21
	Copper (100 mg)	−7.82	0.37
	Manganese (100 mg)	2.64	0.22
	Retinol eq. (100 μg RE)	0.00	0.79
	Vitamin D (100 μg)	0.01	0.79
	α-Tocopherol (100 mg)	−1.05	0.22
	Vitamin K (100 μg)	−0.02	0.16 *
	Vitamin B_1_ (100 mg)	−13.90	0.13 *
	Vitamin B_2_ (100 mg)	−3.41	0.51
	Niacin (100 mg)	−0.22	0.49
	Vitamin B_6_ (100 mg)	−5.37	0.28
	Vitamin B_12_ (100 μg)	0.07	0.72
	Folate (100 μg)	−0.01	0.49
	Pantothenic acid (100 mg)	−1.83	0.14 *
	Alcohol (100 g)	0.14	0.12 *
Food intake			
Vegetables	Pickled green leaves vegetables (100 g)	−0.09	0.58
	Other pickled vegetables (100 g)	−0.09	0.54
	Raw lettuces/cabbage (100 g)	−0.10	0.22
	Green leaves vegetables (100 g)	−0.03	0.52
	Cabbage/Chinese cabbage (100 g)	−0.10	0.11 *
	Carrots/pumpkin (100 g)	−0.10	0.39
	Japanese radish/turnip (100 g)	−0.14	0.12 *
	Other root vegetables (100 g)	−0.07	0.30
	Tomatoes (100 g)	−0.10	0.09 *
Fish and shellfish	Squid/octopus/shrimp/shellfish (100 g)	0.10	0.32
	Small fish with bones (100 g)	0.06	0.60
	Canned tuna (100 g)	−0.15	0.56
	Dried fish/salted fish (100 g)	0.00	0.80
	Oily fish (100 g)	−0.02	0.72
	Lean fish (100 g)	0.01	0.78
	Raw fish (100 g)	0.11	0.08 *
	Grilled fish (100 g)	−0.03	0.51
	Boiled fish (100 g)	0.00	0.80
	Fried fish (100 g)	0.06	0.49
Alcoholic beverage	Sake (100 g)	0.00	0.79
	Beer (100 g)	0.00	0.56
	Shochu (100 g)	0.05	0.11 *
	Whiskey (100 g)	0.05	0.55
	Wine (100 g)	0.03	0.54
Non-alcoholic beverage	Green tea (100 g)	0.01	0.13 *
	Black tea/oolong tea (100 g)	0.02	0.31
	Coffee (100 g)	−0.01	0.51
	Cola drink/soft drink (100 g)	0.00	0.79

* *Q* < 0.20. ^1^ Multiple liner regression model was adjusted for age, sex, body mass index, smoking habit (current habitual smoker or not), drinking habit (current habitual drinker or not), and energy intake. ^2^ Nutrition intake and food intake were adjusted by energy intake using residual method. ^3^ The units used for the calculation of the *β* values are indicated. SFA; saturated fatty acid, MUFA; mono-unsaturated fatty acid, PUFA; poly-unsaturated fatty acid, eq.; equivalent.

### 3.5. Association of Plasma LBP Concentration and Intestinal Microbiota Composition

Multiple regression analysis was conducted using the plasma LBP concentration as the objective variable; composition ratio of intestinal bacteria as the explanatory variable; and age, sex, BMI, smoking habit (current habitual smoker or not), and drinking habit (current habitual drinker or not) as the adjusting factors (Table 5). Results showed that plasma LBP concentration was positively correlated with genus *Prevotella* and its higher groups (phylum Bacteroidetes, class Bacteroidia, order Bacteroidales, family Prevotellaceae), genus *Megamonas*, genus *Streptococcus* and its higher group (family Streptococcaceae). Additionally, and negatively correlated with genus *Roseburia* and its higher groups (class Clostridia, order Clostridiales).

### 3.6. Association of Dietary Factors and Intestinal Microbiota Composition

Multiple regression analysis was conducted with the composition ratio of intestinal bacteria as the objective variable; blood antioxidant concentration, nutrition intake, or food intake as the explanatory variable; and age, sex, BMI, smoking habit (current habitual smoker or not), drinking habit (current habitual drinker or not), and energy intake as adjustment factors (Appendix A). Among the multiple regression analysis results, the dietary factors and intestinal bacteria that showed a significant correlation with plasma LBP concentration in Table 4 and Table 5 were extracted and are shown in Figure 1.

Among the dietary factors that were positively correlated with plasma LBP concentration, alcohol intake, shochu intake, and raw fish intake were positively correlated with intestinal bacteria that were positively correlated with plasma LBP concentration; and negatively correlated with intestinal bacteria that were negatively correlated with plasma LBP concentration. Serum retinol concentration was positively correlated with intestinal bacteria that were positively correlated with plasma LBP concentration. Plasma vitamin C concentration and green tea intake were positively correlated with intestinal bacteria that were negatively correlated with plasma LBP concentration.

Among the dietary factors that were negatively correlated with plasma LBP concentration, serum *β*-cryptoxanthin concentration, serum *β*-carotene concentration, lipid intake, dietary fiber intake, potassium intake, vitamin B1 intake, and pantothenic acid intake were negatively correlated with intestinal bacteria that were positively correlated with plasma LBP concentration; and positively correlated with intestinal bacteria that were negatively correlated with plasma LBP concentration. Japanese radish/turnip intake and tomato intake were negatively correlated with intestinal bacteria that were positively correlated with plasma LBP concentration. Serum total carotenoid concentration, PUFA intake, and vitamin K intake were positively correlated with intestinal bacteria that were negatively correlated with plasma LBP concentration. MUFA intake and cabbage/Chinese cabbage intake did not show a significant correlation with any intestinal bacteria.

## 4. Discussion

In this study, we conducted a cross-sectional study to investigate the association of dietary factors, gut microbiota, and health status with plasma LBP concentration as an index of blood LPS influx in general adult population. Results showed that some of clinical markers, not only for lifestyle-related diseases, but also for what had not been reported to be correlated with blood LPS in previous case-control studies, were also correlated with plasma LBP concentration. Additionally, we observed the relationships between food and nutrition intake in daily life and plasma LBP concentration, those had not been evaluated in previous intervention studies. Furthermore, plasma LBP concentration was correlated with four bacterial genera and their higher groups, and these intestinal bacteria were associated with some of the dietary factors above-mentioned. The individual relationships are discussed below.

### 4.1. Characteristics of the Study Subjects

In Japan, the National Health and Nutrition Survey (NHNS), a survey of the overall health of the population, is conducted regularly by the Ministry of Health, Labour and Welfare. The results of the NHNS conducted in 2017 [30], the same year as this study, showed a habitual smoking rate of 29.3% for male and 7.2% for female, similar to the habitual smoking rate of the participants in this study for both sexes (Table 1). The habitual drinking rate in the NHNS was 53.5% for male and 21.7% for female, similar to this study, with male having a higher habitual drinking rate (Table 1), although the habitual drinking rates in the present study were slightly higher for both sexes than in the NHNS. The health status examined in the NHNS included BMI (23.8 ± 3.4 kg/m^2^ in male and 22.6 ± 3.7 kg/m^2^ in female), HbA1c (5.9 ± 0.8% in male and 5.8 ± 0. 6% in female), total cholesterol (199 ± 36 mg/dL in male and 210 ± 36 mg/dL in female), HDL-c (56 ± 15 mg/dL in male and 67 ± 16 mg/dL in female) and LDL-c (117 ± 32 mg/dL in male and 120 ± 31 mg/dL in female) were similar to our study population (Table 1). Systolic blood pressure in the NHNS was 135 ± 18 mmHg in male and 117 ± 12 mmHg in female, and diastolic blood pressure was 82 ± 11 mmHg in male and 75 ± 10 mmHg in female, with slightly lower values in both sexes in our study population (Table 1), but the results were similar to the NHNS in that male had higher blood pressure than female. The results of the dietary survey of the NHNS showed that intakes of the main nutrients—energy (2134 ± 583 kcal/day in male and 1720 ± 468 kcal/day in female), protein (77 ± 24 g/day in male and 65 ± 21 g/day in female) and carbohydrate (284 ± 86 g/day in male and 233 ± 69 g/day in female)—were at the same level as in our study (Appendix A). The lipid intake of the NHNS subjects was 64 ± 26 g/day for males and 55 ± 23 g/day for females, with a slightly lower lipid intake in both sexes in our study (Appendix A), but the results were similar to the NHNS in that male consumed more fat than female. In a previous report, the major gut bacteria of the Japanese population at the phylum level were Firmicutes, Bacteroidetes, Actinobacteria, and Proteobacteria, accounting for approximately less than 60%, less than 20%, just over 20%, and 2% of the total gut microbiota, respectively [31]. The proportions of these gut bacteria in our study subjects were 62%, 21%, 13% and 3%, respectively (Appendix A), which is very similar to the previous report. Based on the above, it can be assumed that the subjects in this study were not obviously different from the general adult population in Japan in terms of their main health status, nutritional status and the composition of their main gut microbiota.

### 4.2. Health Statuses Correlated with Plasma LBP Concentration

Previous research reported that blood LPS concentration or blood LBP concentration was correlated with obesity [15,17,18], glucose metabolism [15,17], lipid metabolism [15,17,18], vascular function [15,17,32,33], liver function [34,35], and inflammatory conditions [15,17]. The present study also found a significant relationship between plasma LBP concentration and these variables. This study also found that plasma LBP concentration was correlated with muscle mass, blood renin activity, blood cortisol concentration, blood free T4 concentration, and serum iron.

A decrease in muscle mass increases the risk of falls, need for nursing care, risk of chronic diseases, and risk of all-cause mortality in old age [36], so maintaining muscle mass is important in the elderly. In addition, a gradual decline in muscle mass among Japanese people from their late 30s has been reported [37], highlighting the importance of identifying the causes of muscle weakness and maintaining muscle strength not only in the elderly but also in young people. Evidence indicates that sepsis [38] and administration of LPS in animal models [39] induce muscle atrophy. However, epidemiological studies have only reported a correlation between blood LBP concentration and decreased skeletal muscle density in elderly men [16]. The present study included both men and women across a wide age range, and a negative correlation was observed between plasma LBP concentration and muscle mass, independent of age or sex. This suggests that blood LPS influx may results in a decrease in muscle mass from a young age.

Previous research reported that a higher blood LBP concentration in patients with cirrhosis and ascites resulted in increased blood renin activity, and that the administration of the antibacterial drug norfloxacin decreased blood LBP concentration and renin activity [40]. However, the relationship between LPS exposure and blood renin activity in the general adult population was unclear. In the present study, a significant positive correlation was observed between plasma LBP concentration and plasma renin activity, suggesting that LPS exposure is also associated with increased plasma renin activity in the general adult population. Renin is well known to constrict blood vessels and regulate blood pressure via the renin-angiotensin system [41]. Previous research and the present study have shown that blood LBP concentration [17] or blood LPS concentration [32] were positively correlated with blood pressure, and the activation of the renin-angiotensin system resulting from exposure to LPS may partly explain this.

In the present study, plasma LBP concentration was significantly positively correlated with blood cortisol concentration. Previous research showed in an in vitro study that LPS directly acts on adrenal cells and promote cortisol secretion [42]. Additionally, intravenous administration of LPS in humans increased blood cortisol concentration [43,44], suggesting that LPS promotes cortisol secretion. However, to date, the relationship between cortisol concentration and plasma LBP concentration in the general adult population remains unclear. Cortisol has been shown to contribute to increased blood pressure through the regulation of capillary function [45], and like renin, cortisol may be involved in blood pressure elevation upon exposure to LPS. Cortisol is also a well-known stress response hormone and has been found to be associated with depression [46]. In the present study, the score of Center for Epidemiological Studies Depression (CES-D) was used to evaluate depression, but no significant correlation was found between the total CES-D score and blood LBP concentration. Participants in the present study had a total CES-D score of 14 even in the third quartile. In CES-D, a score of 16 or higher is defined as a depressive state, so at least three quarters of the participants in this study were considered to be in a non-depressive state. Therefore, the present study was not suitable for evaluating the correlation between blood LBP concentration and depression. The correlation between increased plasma LBP concentration and increased blood cortisol level-mediated worsening of the depressive state needs to be re-examined in cross-sectional studies involving subjects with different depressive states.

It has been previously reported that sepsis induces apoptosis of thyroid epithelial cells, decreased thyroid function, and decreased free T4, which is a thyroid hormone [47,48]. However, the effect of metabolic endotoxemia on thyroid function has not been elucidated. In the present study, we showed for the first time that there was a positive correlation between plasma LBP concentration and free T4 concentration. Free T4 secretion from the thyroid gland is controlled by the hypothalamic–pituitary–thyroid axis (HPT axis), which is affected by cortisol [49]. However, in the present study, no significant correlation was found between thyrotropin, which constitutes the HPT axis, and plasma LBP concentration. Therefore, it is possible that LPS exposure is involved in the regulation of free T4 secretion through a pathway that does not involve the HPT axis.

We observed negative correlations between plasma LBP concentration and serum iron. Iron in the body is distributed in the liver/spleen, red blood cells, and blood; ferritin, hemoglobin, and serum iron serve as their respective indices. Among these, plasma LBP concentration was significantly correlated only with serum iron. Iron carried in the blood includes iron bound to transferrin that is produced by the liver and iron that is not, and the serum iron measured in the present study refers the former. Previous research has reported that LPS stimulation suppresses transferrin secretion from liver cells [50], suggesting that the negative correlation between plasma LBP concentration and serum iron that was observed in the present study may be related to the suppression of transferrin production. Iron bound to transferrin plays an important role in neuronal metabolism in the brain [51]. Low serum iron has been reported in patients with Alzheimer’s disease compared to healthy subjects [52], and low serum iron that is correlated with plasma LBP concentration may lead to decreased brain function. In the present study, the score of Mini-Mental state examination (MMSE) was used to evaluate cognitive function, although no significant correlations were found between the plasma LBP concentration and MMSE score. The total MMSE score of participants in the present study was 29 even in the first quartile. A MMSE score of 26 or less is interpreted as mild dementia, indicating that at least three-fourths of the participants in the present study did not experience any decline in cognitive function. Therefore, the present study was not suitable for evaluating the relationship between plasma LBP concentration and cognitive function. The relationship between elevated LBP concentration and cognitive decline needs to be re-examined in cross-sectional studies involving study subjects with different cognitive functions.

### 4.3. Dietary Factors Correlated to Plasma LBP Concentration

Here, we discuss the present study’s consistency with previous research and the mechanism of action of the dietary factors that were found to be associated with plasma LBP concentration. The mechanism mediated by intestinal bacteria is described in the section “Relationship between dietary factors and intestinal bacteria” below.

The following dietary factors were found to be correlated with reduced exposure to LPS: dietary fiber [1], MUFA [53], PUFA [53], total carotenoids [54], and *β*-carotene [54]. Our findings are consistent with these results. Additionally, the present study found that plasma LBP concentration was negatively correlated with cabbage/Chinese cabbage intake, Japanese radish/turnip intake, tomato intake, potassium intake, vitamin B1 intake, pantothenic acid intake, vitamin K intake, lipid intake, and serum *β*-cryptoxanthin concentration. 

Cabbage/Chinese cabbage and Japanese radish/turnips are cruciferous vegetables. In a previous study, we reported that extracts of broccoli sprouts, a type of cruciferous vegetable, improved gut microbiota in mice that were fed a high-fat diet, thereby reducing blood LPS concentration [55]. Furthermore, in mice, isothiocyanates that derived from cruciferous vegetables were shown to normalize the intestinal barrier function [56]. Ingestion of cruciferous vegetables may have suppressed the influx of LPS into the blood through this mechanism. 

Lycopene, which is one of the carotenoids characteristically contained in tomatoes, has been reported to suppress intestinal inflammation and lower blood LPS concentration in mice [57]. Therefore, the correlation between tomato intake and plasma LBP concentration observed in the present study may reflect the efficacy of lycopene. However, in the present study, there was no significant correlation between serum lycopene concentration and plasma LBP concentration. Evidence indicates that lycopene is considerably attenuated by digestive juices [58], so there may have been a discrepancy between the action of lycopene in the intestinal tract and the serum lycopene concentration. Additionally, a comprehensive analysis of the components contained in 25 varieties of tomatoes using liquid chromatography tandem-mass spectrometry resulted in the detection of 7118 peaks, of which 1577 were annotated [59]. Many of these are components with anti-inflammatory effects [60]. Therefore, it is thought that components other than lycopene contained in tomatoes may also contribute to the maintenance of the intestinal barrier function and the suppression of metabolic endotoxemia. 

Fat promotes LPS influx, although its effect on blood LPS concentration differs depending on its origin and composition [53]. MUFA have the potential to reduce blood LPS concentration. Results of participant diet analysis showed that the composition ratio of MUFA was the highest among all the fatty acids ingested (Appendix A). Therefore, the effect of this may have been a negative correlation between fat intake and plasma LBP concentration. 

*β*-cryptoxanthin is a carotenoid that is contained in mandarin oranges, paprika, and persimmons. Blood concentration of *β*-cryptoxanthin have been reported to show negative correlations with severity of NAFLD [61], HOMA-IR [62], and BMI [63] in humans. *β*-cryptoxanthin concentration has also been reported to be negatively correlated with the lactulose:mannitol (L/M) ratio, which is an index of intestinal barrier function [64]. Provitamin A carotenoids, including *β*-cryptoxanthin, are cleaved in the intestine by *β*-carotene oxygenase 1, which results in the formation of vitamin A [65]. Vitamin A has been suggested to contribute to the maintenance of the intestinal barrier function [66]. Therefore, *β*-cryptoxanthin may have affected plasma LBP concentration through its provitamin A properties.

Potassium intake or pantothenic acid intake has been suggested to be contribute the reduction of intestinal inflammation [67] or the maintenance of intestinal barrier function [68], respectively. There is limited evidence on vitamin B1 and the intestinal barrier function. However, vitamin B1 deficiency was found to cause sepsis-like conditions [69], suggesting that vitamin B1 may contribute to maintaining the intestinal barrier function. The association of vitamin K and intestinal permeability, barrier function, and inflammation has not been previously reported and further research is necessary to investigate this relationship. 

Alcohol is associated with increased LPS exposure [70], and in the present study a significantly positive correlation was found between the intake of alcohol and shochu, which is a type of Japanese distilled liquor, and plasma LBP concentration. Furthermore, our study found that plasma LBP concentration was positively correlated with green tea intake, raw fish intake, plasma vitamin C concentration, and serum retinol concentration. 

Catechin, which is a flavonoid contained in green tea, may be beneficial in human health through its antioxidant and anti-inflammatory effects [71]. Green tea extract suppresses metabolic endotoxemia in mice [72,73]. This suggests that it is unlikely that green tea itself increases blood LPS concentration. Meanwhile, epidemiological studies have reported that people with a dietary habit of consuming both green tea and alcohol have an increased risk of esophageal cancer [74] and colorectal cancer [75]. In other words, the ingestion of green tea and alcohol may damage the epithelium of the gastrointestinal tract, which may reduce the intestinal barrier function and promote LPS influx. 

BDHQ, which was used in the dietary survey of the present study, investigated three methods of cooking seafood (raw, grilled, boiled), and only raw fish was found to have a significant correlation with plasma LBP concentration. Therefore, it is postulated that eating raw fish, not the fish itself, increases plasma LBP concentration. Raw fish consumption is the cause of food poisoning caused by *Vibrio parahaemolyticus*. PCR testing detected *Vibrio parahaemolyticus* at a frequency of 13–97% in commercially available seafood [76]. Therefore, it is presumed that consumption of raw fish leads to ingestion of *Vibrio parahaemolyticus* toxin in trace amounts, even if it does not lead to food poisoning. The *Vibrio parahaemolyticus* toxin has been reported to damage intestinal epithelial cells [77], so its ingestion may contribute to increased intestinal permeability and blood LPS influx. It is also known that preserving raw fish causes histamine production by the bacteria present on the fish, causing health hazards to humans if the fish is eaten (i.e., histamine fish poisoning). It has been confirmed that histamine content increases during storage in fish eaten in Japan [78]. Oral histamine has been suggested to increase intestinal permeability [79]. Therefore, it is possible that histamine fish poisoning is one of the reasons for the positive correlation between raw fish intake and plasma LBP concentration. Another possibility is that Japanese people with a diet pattern that consumes large quantities of alcohol often consume fish dishes [80,81], so it should be noted that the association between raw fish intake and plasma LBP concentration may be a pseudo-correlation that reflects the amount of alcohol consumed. 

Vitamin C [82] and vitamin A [83] maintain the intestinal barrier function, and we consider that it is unlikely that these directly contributed to the increase in blood LPS concentration. It has been reported that vitamin C intake consist of 10–11% from green tea, and vitamin A intake consist of 30–41% from fish/shellfish in Japanese [84]. Thus, it is possible that the blood concentrations of these two components may indirectly correlate with the plasma LBP concentration due to the association with green tea or raw fish intake. 

### 4.4. Correlation between Plasma LBP Concentration and Intestinal Bacteria

In the present study, plasma LBP concentration was positively correlated with genus *Prevotella* and its higher groups (phylum Bacteroidetes, class Bacteroidia, order Bacteroidales, family Prevotellaceae), genus *Megamonas*, genus *Streptococcus* and its higher group (family Streptococcaceae); and negatively correlated with genus *Roseburia* and its higher groups (class Clostridia, order Clostridiales). 

The correlation between the composition ratio of *Prevotella* and blood LPS concentration was reported in patients with type 2 diabetes [85,86]. *Prevotella* produces succinate as a result of glucose metabolism [87]. Succinate from intestinal bacteria supports the growth of the pathogenic bacteria *Salmonella* serovar Typhimurium [88] and *Clostridium difficile* [89]. Furthermore, succinate induces colitis and promotes colonic fibrosis through succinate receptors [90]. *Prevotella* has also been reported to utilize intestinal mucin [91,92] and increases in mucin-utilizing intestinal bacteria cause disruption of the mucin layer and intestinal inflammation [93]. Therefore, in terms of its nutrient requirements, *Prevotella* may also disrupt the intestinal barrier. 

*Megamonas* was first detected in human feces in 2008 [94], and little is known about its behavior in the human intestinal tract. However, a study comparing the gut microbiota of urban and rural elderly found that gut microbiota composition in the urban elderly had a higher proportion of *Megamonas* and higher fecal LPS concentration [95]. Furthermore, the administration of Kampo medicine (Gegen Qinlian decoction) to patients with colorectal cancer caused a decrease of *Megamonas* among gut microbiota and an improvement in intestinal barrier function [96]. These findings suggest that *Megamonas* is involved in the influx of LPS in the intestinal tract, although the underlying mechanism has not been elucidated. 

*Streptococcus* is a Gram-positive coccus to which many pathogenic bacteria belong. Patients with alcoholic liver disease have a higher composition ratio of *Streptococcus* in the fecal flora, intestinal permeability, and blood LBP concentration compared to healthy subjects [97]. *Streptococcus* is a bacterium that is abundant in the oral cavity; since the composition ratio in the intestinal tract increases in patients treated with proton pump inhibitors [98], it is thought that *Streptococcus* in the oral cavity reaches the intestinal tract due to the decreased digestive capacity of the stomach. This influx of *Streptococcus* has also been suggested to induce an inflammatory response in the colon [99]. 

The gut microbiota composition in patients with type 1 diabetes with high intestinal permeability and high blood LPS concentration is characterized by a decreased proportion of *Roseburia* [100]. *Roseburia* has been shown to strengthen the intestinal barrier function in mice through the production of butyrate and to suppress LPS influx [101]. Regarding the balance of intestinal bacteria at the order level, in the present study, a negative correlation was found between the composition ratio of order Clostridiales and plasma LBP concentration, and a positive correlation was found between the composition ratio of order Bacteroidales and plasma LBP concentration. A previous study comparing gut microbiota of rural and urban children in Thailand found that rural children had the following characteristics [102]: (1) a high composition ratio of order Clostridiales, a low composition ratio of order Bacteroidales; (2) presence of genes involved in the metabolism of plant-derived components and butyric acid production; and (3) butyric acid concentration in feces was high. As mentioned above, butyric acid enhances the barrier function of the intestinal tract. Therefore, order Clostridiales is considered to be a group of bacteria that contribute to the suppression of LPS influx into the blood via butyrate production. We postulated that the balance of intestinal bacteria at the order level observed in the present study has important significance in controlling the influx of LPS into the blood. 

### 4.5. Relationship between Dietary Factors and Intestinal Bacteria

In the present study, we showed a positive correlation between serum retinol concentration and order Bacteroidales. In an interventional study in which autistic children were given supplements that contained vitamin A, vitamin A intake increased the composition ratio of Bacteroidales [103]. However, there is little knowledge about the mechanism by which vitamin A increases Bacteroidales, and further investigation is needed. 

In the present study, shochu and alcohol intake were associated with an increase in genus *Prevotella* and genus *Megamonas*, and a decrease in order Clostridiales. Previous research has also reported that alcohol intake was correlated with an increase in the order Bacteroidales, which is a higher classification of genus *Prevotella* [104]; increase in the genus *Megamonas* [105]; and decrease in order Clostridiales [104]. A possible mechanism is that there are differences in alcohol tolerance among these bacteria. Oral bacteria such as *Prevotella* possess alcohol dehydrogenase and have been reported to acquire alcohol tolerance by metabolizing alcohol to produce acetaldehyde [106]. Meanwhile, *Clostridium* belonging to Clostridiales has been reported to have low alcohol tolerance due to weak alcohol dehydrogenase activity [107]. Based on these reports, we postulate that alcohol intake exerts a selective pressure on intestinal bacteria with low alcohol tolerance, which may affect the composition ratio of the gut microbiota. 

The intake of raw fish dishes was correlated with an increase in the order Bacteroidales and a decrease in the genus *Roseburia*. Since few countries have a custom of eating raw fish, there are no reports identifying the relationships between raw fish dishes and gut microbiota. One study found that humans with a diet consisting mainly of fish had a high composition ratio of family Bacteroidetes in the gut microbiota and low composition ratio of *Roseburia* [108]. However, people whose diet mainly includes fish also consume large quantities of protein foods, refined carbohydrates, vegetables, fruit, juice and sweetened beverages, kid’s meals and snacks and sweets, so the relationship between changes in gut microbiota and fish intake is unclear. In the present study, the pattern of intestinal bacteria associated with the intake of raw fish was similar to that associated with the intake of shochu and alcohol. As mentioned above, Japanese people with a diet pattern that consumes large quantities of alcohol often consume fish dishes. Therefore, there is a need to pay attention to the possibility of a pseudo-correlation that reflects alcohol intake as a background factor for the correlation between raw fish intake and order Bacteroidales as well as genus *Roseburia*. 

Serum *β*-carotene and serum *β*-cryptoxanthin concentrations were both negatively correlated with genus *Streptococcus* and positively correlated with order Clostridiales. There was also a negative correlation between *β*-carotene and genus *Megamonas*. However, no previous reports were found regarding the mechanism by which *β*-cryptoxanthin and *β*-carotene regulate these bacteria. These carotenoids are abundantly contained in vegetables and fruits. As will be described later, dietary fiber intake that is contained in vegetable and fruit intake are reported to be associated with these intestinal bacteria. Therefore, it is postulated that the correlation between *β*-cryptoxanthin and *β*-carotene and intestinal bacteria may be a pseudo-correlation that reflects dietary fiber intake. However, there is an interesting report regarding order Clostridiales. In a previous report comparing the intestinal microbiota of normal subjects and atherosclerosis patients, it was shown that a group of bacteria including Clostridia living in the intestines of normal subjects has phytoene dehydrogenase and may produce *β*-carotene from phytoene in the intestines [109]. In this paper, it is also shown that the blood *β*-carotene concentration of healthy individuals is actually higher than that of atherosclerotic patients, suggesting that carotenoids produced by intestinal bacteria may be absorbed from the intestinal tract and help maintain health. In the present study, *β*-carotene intake that was calculated from BDHQ was not correlated with plasma LBP concentration, raising questions about the origin of serum *β*-carotene. One possible explanation is that bacteria belonging to the order Clostridiales may produce *β*-carotene. 

Potassium and vitamin K were both positively correlated with order Clostridiales. However, the mechanism by which potassium regulates Clostridiales has not been previously reported. Potassium is abundant in vegetables, so it is possible that a pseudo-correlation that was mediated by dietary fiber was observed, similar to *β*-cryptoxanthin and *β*-carotene. Meanwhile, vitamin K has been reported to support the growth of *Faecalibacterium* belonging to the order Clostridiales [110]. Additionally, in this study, a negative correlation was found between potassium intake and genus *Streptococcus*. High concentrations of potassium have been reported to suppress the expression of the pyruvate transporter required for the survival of *Streptococcus mutans* [111], suggesting that potassium intake inhibited the survival of *Streptococcus*. 

Vitamin B1 was negatively correlated with the composition ratios of genus *Prevotella* and genus *Megamonas*, and positively correlated with the composition ratios of order Clostridiales. To the best of our knowledge, there are no reports demonstrating the relationship between vitamin B1 and *Prevotella* and *Megamonas*. On the other hand, it has been reported that vitamin B1 plays an important role in the growth of Clostridiales bacteria [112]. 

Pantothenic acid was positively correlated with class Clostridia and order Clostridiales; and negatively correlated with family Prevotellaceae. Pantothenic acid is a component of coenzyme A (CoA), which is essential for energy acquisition by the glycolytic system, not only in order Clostridiales. CoA is required for the production of butyric acid by bacteria belonging to Clostridiales inhabiting the human intestinal tract [113]. It has been suggested that bacteria belonging to Clostridiales colonize and survive in intestinal mucin [114]. Butyric acid is known to promote mucin production from intestinal epithelial cells and contributes to the adhesion of bacteria to the mucin layer [115]. It is inferred from the above that one of the factors that may have contributed to the positive correlation of pantothenic acid with order Clostridiales was its contribution to the settlement of bacteria belonging to Clostridiales in the intestinal tract through the promotion of butyric acid and mucin production. Regarding the relationship between pantothenic acid and Prevotellaceae, one report indicates that pantothenic acid intake increases the composition ratio of *Prevotella*, which belongs to Prevotellaceae, among intestinal bacteria [116]. However, the underlying mechanisms is unclear. 

Dietary fiber was negatively correlated with the composition ratios of order Bacteroidales and genus *Streptococcus*; and positively correlated with the composition ratio of order Clostridiales. Bacteria belonging to the Order Bacteroidales, including *Prevotella*, are characterized by their ability to utilize with both dietary fiber [117] and mucin [91,92]. It has been reported that such bacteria express mucin-degrading genes when host dietary fiber intake is reduced, then switch nutrient sources, and increase the composition ratio [93]. Therefore, it is postulated that high dietary fiber intake be related to the decrease in the composition ratio of order Bacteroidales. *Streptococcus* is thought to flow into the intestinal tract when the digestive capacity of the stomach declines, as mentioned above. Dietary fiber intake increases the gastric residence time of food [118]. *Streptococcus* has low gastric acid resistance, and its viability may significantly decrease with prolonged residence time in the stomach [119]. These reports suggest that dietary fiber intake inhibits the survival of *Streptococcus* by retaining it in the stomach, contributing to the decrease in its composition ratio in the intestinal tract. Bacteria belonging to the order Clostridiales have been reported to proliferate by utilizing dietary fiber [120,121,122], so it is thought that a positive correlation was observed between dietary fiber intake and order Clostridiales. 

Fat intake was negatively correlated with family Prevotellaceae and genus *Megamonas*, and positively correlated with order Clostridiales. An epidemiological study conducted on Leyte Island showed that the lipid ratio in energy intake was negatively correlated with the composition ratio of family Prevotellaceae and positively correlated with the order Clostridiales [123]. Additionally, animal fat intake has been reported to decrease the composition ratio of *Megamonas* [124]. However, the mechanism by which lipids regulate the composition ratio of these bacteria is unknown.

PUFA intake was positively correlated with order Clostridiales. PUFAs may have antibacterial activity against specific bacteria and may inhibit biofilm formation [125], although Clostridiales may metabolize PUFAs [126], avoiding their antibacterial effects [125]. Therefore, the order Clostridiales may be more likely to survive in the intestines of humans with high PUFA intake, and composition ratios may be higher in the gut microbiota.

Japanese radish/turnip intake was negatively correlated with the genus *Streptococcus*. It has been reported that the 5-methylsulfinyl-1-(4-methylsulfinyl-but-3-enyl)-pent-4-enylidene]-sulfamic acid in radishes has antibacterial activity against *Streptococcus* [127]. Isothiocyanate, which is abundant in cruciferous vegetables such as Japanese radishes and turnips, also has an antibacterial effect on *Streptococcus* [128].

Tomato intake was negatively correlated with genus *Streptococcus*. However, no previous reports have been found on the association of tomatoes and their characteristic component, lycopene, with *Streptococcus*, and further verification is needed to understand the mechanism.

We postulate that a mechanism that is mediated by intestinal bacteria underpins the correlation between dietary factors and plasma LBP concentration observed in the present study may be. However, many correlations have been observed that cannot be fully explained by previous reports. Clarifying the relationship between dietary factors and gut microbiota requires verification experiments using experimental models that reflect the intestinal ecosystem. 

### 4.6. Limitations

There are several limitations in the present study. First, we comprehensively analyzed the correlations of multiple items such as dietary factors, intestinal bacteria, and health status. Therefore, although we adjusted for false discovery rates with the Storey method, it is possible that some of the significant correlations observed in this study may have been coincidental. Second, this study involved residents of Iwaki district of Hirosaki city, Aomori prefecture, and further research is needed to establish whether these results can be extrapolated to other regions and races. Third, the causality of the correlations obtained here cannot be inferred because the present study is a cross-sectional study. These findings should be verified by longitudinal studies and intervention studies. Fourth, although the BDHQ used for the diet survey in the present study has been fully validated and has been used in other cross-sectional and cohort studies, inherent misclassification remains a possibility due to the self-reporting nature of the study. Finally, LBP is also produced by differentiated adipocytes in an LPS-independent manner [129] and plasma LBP concentration does not only reflect blood LPS influx. Therefore, it is necessary to confirm the reproducibility of the results obtained in this study in a study in which the plasma LPS concentration itself is used as an indicator of exposure to LPS.

## 5. Conclusions

In this study, we found that plasma LBP concentration in a general adult population was correlated not only with previously known factors such as obesity, glucose metabolism, vascular function, and liver function; but also with physical function, renal function, adrenal cortical function, thyroid function, and iron metabolism. Maintaining low plasma LBP concentrations may be associated with the maintenance of a wider range of factors of health status than was previously assumed. Additionally, the present study suggested that the food and nutrition intake in daily life associated with plasma LBP concentration, and that the intestinal bacteria are involved as a background factor. These findings are expected to be useful for the provision dietary guidance to suppress the influx of LPS into the blood and maintain health.

## Figures and Tables

**Figure 1 metabolites-13-00250-f001:**
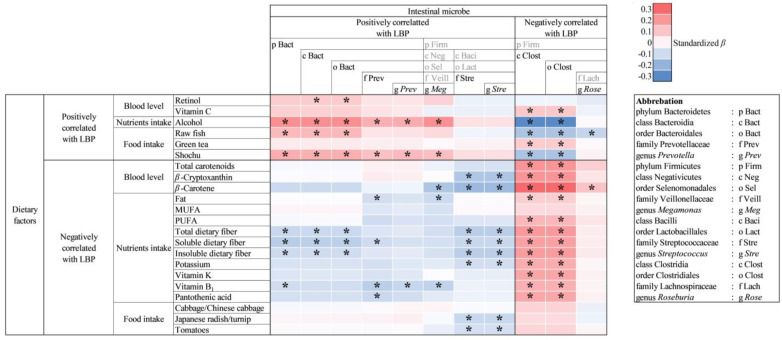
Heatmap showing the relationship between dietary factors and intestinal microbiota which were correlated with plasma LBP concentration. Each intestinal bacterium was listed according to bacterial classification, and bacteria not used in the analysis were indicated by gray letters. Standardized *β* was shown in red (positive correlation) or blue (negative correlation). False discovery rates were evaluated by Storey method. * *Q* < 0.20.

**Table 1 metabolites-13-00250-t001:** Participant characteristics (n = 896).

	All Subjects			Male			Female		
Variables	Mean ± SD or n (%)	Median	IQR	Mean ± SD or n (%)	Median	IQR	Mean ± SD or n (%)	Median	IQR
Sex (male/female)	375 (41.9)/521 (58.1)	NA	NA	NA	NA	NA	NA	NA	NA
Age (years)	54.7 ± 15.0	56.0	43.0–67.0	54.2 ± 15.3	55.0	41.0–67.0	55.0 ± 14.9	57.0	43.0–66.0
Current smoker	128 (14.3)	NA	NA	91 (24.3)	NA	NA	37 (7.1)	NA	NA
Current drinker	435 (48.5)	NA	NA	261 (69.6)	NA	NA	174 (33.4)	NA	NA
Body mass index (kg/m^2^)	23.1 ± 3.6	22.8	20.5–25.1	23.9 ± 3.4	23.5	21.6–25.8	22.5 ± 3.6	22.2	19.8–24.4
Abdominal circumference (cm)	85.0 ± 10.0	84.7	78.6–90.6	89.0 ± 9.3	87.8	83.2–93.6	82.2 ± 9.5	81.6	75.4–87.8
Systolic blood pressure (mmHg)	124 ± 18	122	111–134	127 ± 18	125	114–137	121 ± 17	120	109–131
Diastolic blood pressure (mmHg)	72 ± 12	71	63–79	74 ± 12	73	66–82	70 ± 11	69	62–78
baPWV (cm/s)	1500 ± 382	1440	1220–1690	1567 ± 404	1498	1285–1730	1445 ± 358	1386	1170–1668
HOMA-IR	1.32 ± 1.27	1.03	0.78–1.50	1.41 ± 1.72	1.03	0.74–1.52	1.25 ± 0.81	1.02	0.80–1.49
Blood glucose (mg/dL)	95.1 ± 16.2	91.5	86.0–99.0	98.6 ± 19.3	94.0	88.0–103.0	92.5 ± 12.8	90.0	84.0–97.0
HbA1c (%)	5.68 ± 0.58	5.60	5.40–5.80	5.71 ± 0.66	5.60	5.40–5.80	5.66 ± 0.52	5.60	5.40–5.80
Blood insulin (μU/mL)	5.42 ± 5.05	4.50	3.50–6.20	5.59 ± 7.14	4.40	3.20–6.25	5.29 ± 2.68	4.60	3.60–6.20
Triglyceride (mg/dL)	100 ± 73	80	58–116	127 ± 94	96	71–150	81 ± 44	70	53–97
Total cholesterol (mg/dL)	207 ± 34	204	185–231	204 ± 33	202	182–227	210 ± 34	207	187–234
HDL-cholesterol (mg/dL)	65 ± 17	63	53–76	59 ± 16	55	48–68	70 ± 16	69	58–79
LDL-cholesterol (mg/dL)	116 ± 28	115	97–133	114 ± 28	113	95–132	117 ± 29	116	98–134

SD, standard deviation; IQR, interquartile range; baPWV, brachial-ankle pulse wave velocity; HOMA-IR, homeostasis model assessment-insulin resistance; HbA1c, hemoglobin A1c; HDL, high-density lipoprotein; LDL, low-density lipoprotein; NA, not analyzed.

**Table 2 metabolites-13-00250-t002:** Distribution of plasma lipopolysaccharide-binding protein (LBP) concentration (μg/mL) and its relation to characteristics of participants.

Variable	n	Mean ± SD	Median	IQR	*p*
All subjects	896	5.77 ± 1.68	5.66	4.63–6.70	
Age strata (years)					
20–29	46	5.46 ± 1.85	5.15	4.25–6.81	<0.001 ^1^
30–39	141	5.29 ± 1.73	5.12	4.09–6.17	
40–49	140	5.73 ± 1.69	5.53	4.58–6.52	
50–59	183	5.83 ± 1.65	5.75	4.73–6.81	
60–69	239	5.85 ± 1.62	5.87	4.85–6.74	
70–79	125	6.12 ± 1.59	6.06	4.96–6.99	
≥80	22	6.23 ± 1.62	6.37	5.14–7.37	
BMI strata (kg/m^2^)					
<18.5	62	5.06 ± 1.53	4.84	4.09–5.82	
18.5–24.9	605	5.64 ± 1.67	5.54	4.52–6.57	0.01 (vs. <18.5) ^2^
≥25.0	229	6.29 ± 1.62	6.17	5.24–7.12	<0.001 (vs. other 2 groups) ^2^
Sex					
Male	375	5.93 ± 1.62	5.88	4.95–6.86	0.002 ^3^
Female	521	5.65 ± 1.71	5.48	4.47–6.55	
Smoking habit					
Non-smoker	768	5.72 ± 1.64	5.63	4.57–6.64	0.04 ^3^
Current smoker	128	6.06 ± 1.88	5.91	4.97–6.89	
Drinking habit					
Non-drinker	461	5.77 ± 1.67	5.67	4.67–6.73	0.96 ^3^
Current drinker	435	5.76 ± 1.69	5.66	4.62–6.67	

^1^*p* for trend was evaluated by Jonckheere-Terpstra test. ^2^ Comparison of median values between the three BMI groups were performed with Steel-Dwass test, and *p* values were given for each group comparison. ^3^ Comparison of median values between the two groups were performed with Mann-Whitney *U* test. SD; standard deviation, IQR; interquartile range.

**Table 3 metabolites-13-00250-t003:** Association between plasma LBP concentration and clinical markers or clinical scores ^1^ (Only categories that were significantly associated with plasma LBP concentrations are selected).

Category	Variable	*β*	*Q* Value
Obesity	Abdominal circumference (cm) ^2^	0.02	0.05 *
	Body fat percentage (%)	0.03	0.32
	Body water content (kg)	−0.02	0.21
	Visceral fat level (levels)	0.09	0.03 *
	Basal metabolic rate (kcal/day)	−0.01	0.53
	Basal metabolic rate level (levels)	−0.08	0.03 *
Blood pressure	Systolic blood pressure (mmHg)	0.07	<0.001 *
	Diastolic blood pressure (mmHg)	0.05	0.05 *
Arteriosclerosis	baPWV (cm/s)	0.08	<0.001 *
Glucose metabolism	HOMA-IR	0.18	0.04 *
	Blood glucose (mg/dL)	0.02	0.41
	HbA1c (%)	0.04	0.009 *
	Blood insulin (μU/mL)	0.16	0.04 *
	C peptide (ng/mL)	0.14	0.009 *
	Glycoalbumin (%)	0.02	0.37
Lipid metabolism	Triglyceride (mg/dL)	0.13	0.14 *
	Total cholesterol (mg/dL)	0.01	0.63
	HDL-cholesterol (mg/dL)	−0.12	<0.001 *
	LDL-cholesterol (mg/dL)	0.04	0.40
Physical function	Estimated bone mass (kg)	−0.01	0.63
	Total muscle mass (kg)	−0.01	0.52
	Right leg muscle mass (kg)	−0.02	0.27
	Left leg muscle mass (kg)	−0.02	0.29
	Right arm muscle mass (kg)	−0.02	0.19 *
	Left arm muscle mass (kg)	−0.03	0.13 *
	Trunk muscle mass (kg)	0.00	0.80
	Locomo 25 score (points)	−1.84	0.24
Eyesight	Far-sightedness	−0.05	0.49
	Near-sightedness	−0.02	0.73
Liver function	Albumin (g/dL)	−0.01	0.16 *
	AST (U/L)	0.16	<0.001 *
	ALT (U/L)	0.17	0.02 *
	γ-GTP (U/L)	0.31	<0.001 *
	Total bilirubin (mg/dL)	−0.11	0.07 *
	Total Protein (g/dL)	0.03	<0.001 *
Renal function	Creatinine (mg/dL)	0.02	0.48
	Blood urea nitrogen (mg/dL)	−0.01	0.75
	Plasma renin activity (ng/mL/h)	0.27	0.16 *
	Urine albumin creatinine ratio (mg/gCr)	0.50	0.004 *
Adrenal cortex function	Aldosterone (pg/mL)	−0.01	0.77
	Cortisol (μg/dL)	0.14	0.02 *
Thyroid function	Free thyroxine (ng/dL)	0.05	0.02 *
	Thyroid stimulating hormone (μIU/mL)	−0.07	0.58
Inflammation	Neutrophil (%)	0.11	<0.001 *
	Stab neutrophil (%)	0.44	<0.001 *
	Segmented neutrophil (%)	0.09	<0.001 *
	Lymphocyte (%)	−0.16	<0.001 *
	Monocyte (%)	0.09	0.20
	Eosinophil (%)	−0.67	0.06 *
	Basophil (%)	−0.18	0.10 *
	IgG (mg/dL)	0.06	0.10 *
	IgA (mg/dL)	0.17	0.009 *
	IgM (mg/dL)	0.06	0.52
	Complement C3 (mg/dL)	0.16	<0.001 *
	Complement C4 (mg/dL)	0.25	<0.001 *
	hs-CRP (mg/dL)	1.52	<0.001 *
	IL-6 (pg/mL)	0.43	<0.001 *
Hematological test	White blood cell (cells/μL)	0.16	<0.001 *
	Red blood cell (10^4^ cells/μL)	−0.01	0.54
	Hemoglobin (g/dL)	−0.01	0.54
	Hematocrit (%)	−0.54	0.32
	Mean cell volume (fL)	0.00	0.60
	Mean corpuscular hemoglobin (pg)	0.00	0.80
	Mean cell hemoglobin concentration (g/dL)	0.00	0.46
	Platelet (10^4^ cells/μL)	0.05	0.23
Iron metabolism	Ferritin (ng/mL)	−0.02	0.77
	Serum iron (mmol/L)	−0.14	0.07 *

* *Q* < 0.20. ^1^ Multiple liner regression model was adjusted for age, sex, body mass index, smoking habit (current habitual smoker or not), drinking habit (current habitual drinker or not). ^2^ The units used for the calculation of the *β* values are indicated. baPWV, brachial-ankle pulse wave velocity; HOMA-IR, homeostasis model assessment-insulin resistance; HbA1c, hemoglobin A1c; HDL, high-density lipoprotein; LDL, low-density lipoprotein; AST, aspartate aminotransferase; ALT, alanine aminotransferase; *γ*-GTP, *γ*-glutamyltransferase; Ig, immunoglobulin; hs-CRP, high-sensitivity C-reactive protein; IL, interleukin.

**Table 5 metabolites-13-00250-t005:** Association between plasma LBP concentration and intestinal microbiota composition ^1^.

Variable ^2^	*β*	*Q* Value
phylum Actinobacteria	−0.07	0.65
class Actinobacteria	−0.07	0.65
order Bifidobacteriales	−0.12	0.56
family Bifidobacteriaceae	−0.12	0.56
genus *Bifidobacterium*	−0.12	0.56
class Coriobacteriia	NA	NA
order Coriobacteriales	0.03	0.79
family Coriobacteriaceae	0.03	0.79
genus *Collinsella*	0.01	0.80
phylum Bacteroidetes	0.26	0.16 *
class Bacteroidia	0.25	0.17 *
order Bacteroidales	0.25	0.17 *
family Prevotellaceae	0.30	0.04 *
genus *Prevotella*	0.32	0.04 *
family Bacteroidaceae	−0.17	0.49
genus *Bacteroides*	−0.17	0.49
family Rikenellaceae	−0.31	0.57
genus *Alistipes*	−0.28	0.60
family Porphyromonadaceae	−0.59	0.54
genus *Parabacteroides*	0.06	0.79
phylum Firmicutes	−0.16	0.44
class Clostridia	−0.21	0.17 *
order Clostridiales	−0.21	0.17 *
family Lachnospiraceae	−0.12	0.49
genus *Blautia*	0.03	0.79
genus *Anaerostipes*	−0.20	0.52
genus *Roseburia*	−0.57	0.18 *
genus *Fusicatenibacter*	−0.76	0.26
genus *Lachnospiraceae incertae sedis*	−0.28	0.73
family Ruminococcaceae	−0.17	0.32
genus *Faecalibacterium*	−0.32	0.29
genus *Ruminococcus*	−0.25	0.50
genus *Ruminococcus 2*	0.05	0.76
genus *Gemmiger*	−0.36	0.56
family Clostridiaceae	NA	NA
genus *Clostridium IV*	−0.14	0.68
class Negativicutes	0.50	0.21
order Selenomonadales	0.50	0.21
family Veillonellaceae	0.46	0.26
genus *Megamonas*	0.56	0.1998 *
class Erysipelotrichia	−0.07	0.77
order Erysipelotrichales	−0.07	0.77
family Erysipelotrichaceae	−0.07	0.77
class Bacilli	0.43	0.26
order Lactobacillales	0.45	0.23
family Streptococcaceae	0.64	0.16 *
genus *Streptococcus*	0.65	0.15 *
phylum Proteobacteria	0.14	0.73
class Betaproteobacteria	−1.10	0.37
order Burkholderiales	−1.10	0.37
family Sutterellaceae	−1.06	0.38
family Unclassified	0.56	0.59
genus Unclassified	0.31	0.41

* *Q* < 0.20. ^1^ Multiple liner regression model was adjusted for age, sex, body mass index, smoking habit (current habitual smoker or not), drinking habit (current habitual drinker or not). ^2^ Each intestinal bacterium was listed according to bacterial classification. NA; not analyzed.

## Data Availability

The data are not publicly available due to the ethical concerns. Data are available from the Hirosaki University COI Program Institutional Data Access/Ethics Committee for researchers who meet the criteria for access to the data.

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
