# Peer review of "Association of Plasma Lipopolysaccharide-Binding Protein Concentration with Dietary Factors, Gut Microbiota, and Health Status in the Japanese General Adult Population: A Cross-Sectional Study"

_metabolites, 2023, doi:10.3390/metabo13020250_

Round 1
Reviewer 1 Report
The article presented for review " Association of Plasma Lipopolysaccharide-Linding Protein Concentration with Dietary Factors, Gut Microbiota, and Health Status in the Japanese General Adult Population: A Cross-Sectional Study" deals with an interesting and relevant
topic from a medical perspective.
In their study, the authors assessed dietary and other nutrient intake, gut microbiota, health status, and plasma concentrations of LPS-binding protein (LBB).
The study was conducted on a group of 896 Japanese residents.
In my opinion, the study was planned correctly. The results of the study were presented in the form of 4 tables and 3 figures.
Appropriate advanced statistical methods were used in the statistical analysis.
The authors on the basis of the study reached the right conclusions. Noteworthy is the discussion chapter. The authors have discussed the results of their research in a very thorough and comprehensive form.
The paper cites 123 items of current scientific literature, which indicates an accurate analysis of the research topic.
My doubts are raised by the presentation of the research results in Tables 2 and 3. The authors should consider whether the results can be presented in a shorter and clearer form.
In my opinion, the paper can be accepted for publication in its present form after a simplified presentation of the results in Tables 2 and 3.
Reviewer 2 Report
The manuscript entitled „Association of plasma lipopolysaccharide-binding protein concentration with dietary factors, gut microbiota, and health status in the Japanese general adult population: A cross-sectional study” presents interesting issue, but some problems should be corrected.
Abstract:
Authors should clearly indicate the aim of the study (e.g. “The aim of the study was…”) instead of only what was done (“we measured…”).
The general characteristics of the studied group should be presented (e.g. gender, age, etc.)
Authors should present numeric results of their study accompanied by the results of their statistical analysis (p-Value).
Introduction:
Authors should clearly present the definitions – in the current version of the manuscript even the definition of lipopolysaccharides is improper, as Authors define them in their 1st sentence as “a cell component of gram-negative bacteria”, while in fact they are molecules consisting of a lipid and a polysaccharide (this is a clear definition).
Authors should in this paragraph clearly justify the need to conduct the study, by presenting the current state of knowledge, gaps in existing knowledge and specifying what should be verified.
Lines 71-81 – the project should not be described in this section, but rather in Materials and Methods Section
Lines 88-96 – the objective of the study should not be presented in this section, but rather in Materials and Methods Section
Materials and Methods:
How were participants recruited?
Were participants representative for city, province, or country?
What was the characteristics of participants?
How was “habitual drinker” defined?
For conducted biochemical analysis the specific method should be defined. Authors indicated in supplementary material only the company which conducted measurement (“Measurements were performed by LSI Medience Corp. (Tokyo, Japan)”), while not company, but detailed method is crucial to assess obtained results.
Detailed information about diet history questionnaire are needed – What questions were asked? Was it FFQ or not? Were respondents asked about food products or food groups? How were datat recalculated? Etc. Based on the data presented in Results Section, it may be supposed that only some food items were included – which were included and why were they chosen?
Results:
It should be clearly indicated which data are normally distributed and which are not
Instead of figures, Authors should present tables to be easier to follow by readers
Table 1 – Authors should compare characteristics of male and female participants for all variables
Table 2 – for all variables Authors should define units – e.g. Basal metabolic rate level (kcal?), Visceral fat level (%? Kg?), etc.
Table 3 – what do Authors mean by 100g, 100 mg, 100 ug for nutrients and foods? How did they recalculate them per 100? And why?
Discussion:
Table 3 is not needed here and should be rather presented as graphical abstract
Reviewer 3 Report
The authors measured intake of foods and nutrients, gut microbiota, health status, and plasma lipopolysaccharide-binding protein (LBP) concentration, an exposure indicator to LPS, in 896 residents of the Iwaki district, a rural area of Japan, and each correlation was analyzed.
The manuscript is well written and the results are well presented.
Noticed a typo in the title "Lipopolysaccharide-Linding Protein" should be "Lipopolysaccharide-Binding Protein". Please correct it.
Round 2
Reviewer 2 Report
The manuscript entitled „Association of plasma lipopolysaccharide-binding protein concentration with dietary factors, gut microbiota, and health status in the Japanese general adult population: A cross-sectional study” presents interesting issue, but some problems should be corrected.
Abstract:
Authors should present numeric results of their study (obtained values) accompanied by the results of their statistical analysis (p-Value).
Introduction:
Lines 97-99, lines 108-109 – the methodological aspects should not be described in this section, but rather in Materials and Methods Section
Results:
Table 1 – Authors should compare characteristics of male and female participants for all variables – Authors should conduct statistical analysis and present p-Values (information that something is “obviously higher” is not sufficient)
Table 2 – for all variables Authors should define units – e.g. Basal metabolic rate level, Visceral fat level – what does “levels” mean? It is not defined within Materials and Methods Section
Table 3 – what do Authors mean by 100g, 100 mg, 100 ug for nutrients and foods? How did they recalculate them per 100? And why? It should be explained within Materials and Methods Section
